# Synthesis of Fluorescent, Dumbbell-Shaped Polyurethane Homo- and Heterodendrimers and Their Photophysical Properties

**DOI:** 10.3390/ijms24021662

**Published:** 2023-01-14

**Authors:** Dhruba P. Poudel, Richard T. Taylor

**Affiliations:** Department of Chemistry and Biochemistry, Miami University, Oxford, OH 45056, USA

**Keywords:** polyurethane dendrimers, homodendrimers, heterodendrimers, Janus dendrimers, fluorescence, click reaction, late-stage modification, absorption, emission

## Abstract

Fluorescent dendrimers have wide applications in biomedical and materials science. Here, we report the synthesis of fluorescent polyurethane homodendrimers and Janus dendrimers, which often pose challenges due to the inherent reactivity of isocyanates. Polyurethane dendrons (G1–G3) were synthesized via a convergent method using a one-pot multicomponent Curtius reaction as a crucial step to establish urethane linkages. The alkyne periphery of the G1–G3 dendrons was modified by a copper catalyzed azide–alkyne click reaction (CuAAC) to form fluorescent dendrons. In the reaction of the surfaces functionalized two different dendrons with a difunctional core, a mixture of three dendrimers consisting of two homodendrimers and a Janus dendrimer were obtained. The Janus dendrimer accounted for a higher proportion in the products’ distribution, being as high as 93% for G3. The photophysical properties of Janus dendrimers showed the fluorescence resonance energy transfer (FRET) from one to the other fluorophore of the dendrimer. The FRET observation accompanied by a large Stokes shift make these dendrimers potential candidates for the detection and tracking of interactions between the biomolecules, as well as potential candidates for fluorescence imaging.

## 1. Introduction

Dendrimers, as a pervasive class of macromolecules, have been extensively investigated for over four decades because of their unique monodispersity and multifunctionality. Depending on the types of functionalities present throughout the dendritic wedges, the dendrimers can be classified into two groups. The first group consists of identical functionalities throughout the dendrons, and thus in the dendrimer. These dendrimers are often called ‘homodendrimers.’ The second group consists of unidentical or dissimilar functions in the dendrons, and the dendrimers are classified as heterodendrimers. Such heterodendrimers composed of two different dendritic wedges of different hydrophobicity or hydrophilicity are called “Janus dendrimers (JDs)” [1,2,3,4]. The presence of dissimilar functions in the dendrons enable JDs to combine different properties associated with different wedges. Compared to amphiphilic copolymers [5,6], their dendritic analogs (JDs) offer more tunability at the molecular level, further affecting their self-assembling nature, where the tunning may be performed for peripheral groups, dendron generation, and the density of branching [3].

In a review, Caminade et al. described three distinct methods for the synthesis of JDs, as shown in Figure 1 [1]. The first method is the simplest of all three and consists of reacting two dendrons with complementary functionalities. This method simply joins two different dendrons together at their focal points. Dendrons with semifluorinated benzyl ethers moiety connected together to form an amide linkage in the JD [7], and a Pd-catalyzed coupling of two focal points of dendrimeric wedges to create the JD are typical examples of this method [8]. The second method involves the convergent synthesis of dendrons followed by the controlled attachment of one dendron to core (with even number of functionalities, most often a difunctional) via a dendron’s focal point. The remaining reactive site of core is then reacted with the other dendron to afford a JD. The very first example of all types of JDs synthesized by Hawker et al. in 1993 (poly(benzyl) ether) stands out as an excellent example of creating JDs using the second method [9]. The third method consists of the convergent synthesis of a dendron. The focal point of the dendron is then utilized to grow new branches using a divergent method [10,11]. Moreover, the syntheses of these dendritic structures via self-assembly caused by noncovalent interactions can also be found in the literature [3,12].

Employing these methods, several JDs containing various types of linkages have been reported by many research groups since 1993. Percec et al. elegantly described the sequence-defined Janus glycodendrimers that can be designed to mimic the spatial properties of biological membranes, thereby providing a versatile tool in glycobiology [13,14]. In another study, Janus polyethylene glycol (PEG)-based dendrimers have been reported to perform controlled multi-drug loading and sequential release [15]. Buzzacchera et al. recently reported the synthesis of constitutional isomeric libraries of self-assembling dendrons and JDs employing natural and synthetic phenolic acids and PEGs [16]. These dendrimers can be utilized as important tools for nanomedicine and synthetic cell biology. The ongoing research on the potential applications of JDs in the biomedical field is further supported by the report on amphiphilic spermine–alkyl Janus dendrimers, which self-assemble to form highly ordered crystalline virus assemblies. Such findings can be used for applications in the study of complex biological systems [17]. Moreover, a decade ago, the polymers that are dendronized with self-assembling JDs, possessing one fluorinated and one hydrogenated dendron, were shown to act as reverse thermal actuators [18]. JDs not only have applications in the field of biomedical science but are also widely applicable in material science. These dendritic macromolecules are reported to have suitable properties for making optoelectronic devices. The synthesis of a highly conjugated JD, containing ferrocene in one denron and cyano-groups in the other dendron and synthesized from a convergent multistep, has been reported to show strong donor–acceptor properties, opening a gateway to optical and electronic devices [19]. Amphiphilic JDs sometimes self-assemble to form vesicles, often called dendrimersomes, which offer improved stability and versality over liposomes. In recent years, the synthesis and applications of many dendrimersomes in membrane and imaging science have been reported [20,21,22,23,24].

This study is motivated by these aforementioned reports and our previous research on the synthesis of low-generation blue and yellow fluorescent polyurethane dendrimers (PUDs) [25]. Fluorescent JDs are different from non-fluorescent analogs since they have fluorophores in one of the dendrons or different fluorophores in two different dendrons. They offer promising applicability for labeling or biological entities.

Click chemistry is a commonly used tool in chemistry for the synthesis and functionalization of a wide range of macromolecules. Copper-catalyzed azide–alkyne cycloaddition (CuAAC) is one of the most widely exploited click reactions, following the pioneering studies by Sharpless and Meldal conducted in 2002 [26,27]. Since then, an azide–alkyne click reaction has been universally employed in both convergent and divergent methods for the synthesis of Janus dendrimeric structures [28,29,30,31,32]. As a model, our group has previously reported the use of click chemistry in the synthesis and late-stage modification of polyurethane dendrimers [33].

Expanding on our previously reported research on low-generation fluorescent dendrimers [25], the objectives of the current study are threefold: (a) to demonstrate that the azide–alkyne click reaction is an effective strategy for preparing higher-generation dendrons, (b) to ascertain the utility of forming Janus-type dendrimers by reacting different dendrons with the core in a single step, and (c) to assay the extent of FRET as the generations of dendrimers increase. For this, one-pot multicomponent Curtius rearrangement was employed to furnish a generation-one (G1) dendron. The subsequent coupling of the G1 dendron with a linking group followed by spacer group attachment afforded a G2 dendron, which in turn, afforded a G3 dendron following the same synthetic route. The late-stage modification of the G1–G3 dendrons via CuAAC utilizing two non-fluorescent small compounds (7-diethylamino-3-azidocoumarin **4** and 4-azido-N-ethyl-1,8-naphthalimide **5**) afforded surface-modified dendrons with different fluorescence values. When two different dendrons were allowed to react with a difunctional core, a mixture of three dendrimers was obtained, containing two homodendrimers and a Janus dendrimer. Surprisingly, more than 65% of the product’s distribution comprised a Janus dendrimer. This method of JDs synthesis has not previously been reported.

## 2. Results and Discussion

### 2.1. Synthesis of G1 Dendron

The synthesis of generation-one dendron **3** has been previously reported by our group [25]. In brief, a protecting group-free approach using a one-pot multicomponent reaction was employed, where a mixture of 5-hydroxyisophthalic acid **1**, diphenylphosphoryl azide (DPPA), and triethylamine was heated to generate an isocyanate in situ, which was then trapped by 4-pentyn-1-ol to form phenolic diurethane **2**. The subsequent attachment of 11-bromoundecanol as a spacer group afforded the G1 dendron **3** in a two-step sequence (Figure 2 and Appendix A).

### 2.2. Late-Stage Modification (LSM) of G1 Dendron **3**

Late-stage modification is one of the most powerful synthetic approaches for the synthesis and functionalization of polymeric or dendritic macromolecules, especially when the functional groups in the molecule are sensitive to at the beginning of the synthetic route. To execute the LSM of dendrons, two different azido-compounds, **4** [34] (Appendix A) and **5** [35,36], were synthesized using a previously reported procedure (Appendix A). The LSM of **3** using non-fluorescent **4** under azide–alkyne click conditions yielded blue–fluorescent G1 dendron **7,** as shown in Figure 3. The formation of 1,4-substituted-1,2,3-triazole after a click reaction led to a highly fluorescent PU dendron. In a similar approach, the same dendron was clicked with a different azide-clicking partner **5** (Appendix A ) to furnish a modified dendron **7** (Appendix A). However, the expected fluorescence of **7** was not observed (under irradiation using 365 nm UV light) after azide–alkyne click reaction as reported in the literature [37]. Both of these reactions preceded smoothly at room temperature for 2.5 h, providing excellent product yields. The pure products were obtained by flash chromatography, where the fluorescent dendrons could be easily visualized using a UV lamp during purification. Alternatively, the product can be precipitated in water without requiring chromatography.

### 2.3. Synthesis of G1 Homo- and Janus Dendrimers

With late-stage-modified dendrons **6** and **7** in hand, G1 homo- and Janus dendrimers were synthesized by allowing these two dendrons (1.2 eq each) to react with hexamethylene-1,6-diisocaynate core **8** at room temperature for 20 h (Figure 4 and Appendix A). The reaction was catalyzed by dibutyltin dilaurate (DBTDL), which is a workhorse catalyst in urethane chemistry used to convert an isocyanate into a urethane linkage [38]. The crude reaction mixture was purified by flash chromatography to afford a mixture of two homodendrimers, **9** and **11,** and a Janus dendrimer **10** as yellow solids. Of these three dendrimers, **9** and **10** were found to exhibit a strong fluorescence; however, dendrimer **11** did not fluoresce when illuminated with a 365 nm UV lamp. Moreover, 77% of the product’s total yield was constituted by the blue-fluorescent Janus dendrimer **10**. Due to the presence of long non-polar alkyl chains, these solid dendrimers did not have sharp melting points.

### 2.4. Growth of Higher-Generation Dendrons

The synthesis of dendrons to the G3 dendrons is shown in Figure 5 and Appendix A. The linking group **12** required for the growth of the dendrons was synthesized and reported in previous studies [39]. To prepare G2 dendron **14**, a mixture of G1 dendron **3** and linking group **12** was heated in anhydrous DMF under Curtius reaction conditions for three days to obtain a highly viscous dark red oil as a crude product, which yielded phenolic dendron **13** on chromatographic purification. This reaction was carried out in absence of any catalyst, where longer reaction times tend to provide better yields. The reaction set for about 36 h only had a 42% yield. Moreover, only 1.5 eq of **3** was found to afford the highest product yield. This could be due the fact that the phenolic OH of **12** can compete with the alcoholic OH of **3** during a nucleophilic attack on in situ generated isocyanate, leading to polymeric side products. This required us to utilize less than 2 eq of dendron for this specific reaction. The attachment of the undecanol group at the phenolic position in the next step afforded G2 dendron **14** with an excellent yield (90%).

Next, G2 dendron **14** was heated with same linking group **12** following the same route of reaction to afford G3 phenolic dendron **15**. With an increase in the size (branching) of the dendron, the steric effects also increase, thereby decreasing the yield of the reaction. Accordingly, the yield of **15** was slightly lower (62%) than that of G2 phenolic dendron, which in the next step underwent an S_N_2 reaction with 11-bromoundecanol in presence of potassium carbonate as a base to afford G3 dendron **16** with an 80% yield. Both G2 and G3 dendrons can be easily purified by rapidly eluting them through a small silica column. Physically, these dendrons are highly viscous at room temperature, and this viscosity decreases when heat is gently applied, emonstrating a sol–gel behavior. This behavior can be attributed to the presence of numerous urethane (NHCOO) linkages, which could be associated with the formation of noncovalent interactions such as hydrogen bonding. Such gel-like compounds could have potential applications in material and biological sciences.

### 2.5. Late-Stage Modification of G2–G3 Dendrons

We utilized LSM approach to further functionalize the periphery of G2–G3 dendrons using CuAAC, as previously shown in G1 dendrons and dendrimers (Figure 6 and Appendix A). Firstly, the peripheral terminal alkyne groups were modified into 1,2,3-triazoles using azidocoumarin **4** as an azide-clicking partner. The reaction was fast and efficient and was performed at room temperature to produce a highly fluorescent (blue) dendron **18** with a 96% yield. Under similar conditions and using the same set of catalysts (copper sulfate and sodium ascorbate), a different G2 dendron **19** was produced with an 87% yield, employing azidonaphthalimide as the azide-clicking partner in this case. The expected fluorescence, however, was not observed (when illuminated with 365 nm UV lamp) in this molecule, which is contrary to the reported literature.

After the successful surface functionalization of G2 dendrons using azide–alkyne click chemistry, we began modifying G3 dendron **16** using azides **4** and **5**. For this, G3 dendron and azidocoumarin **4** were dissolved in THF, to which a brown solution of copper sulfate and sodium ascorbate in minimum amount of water was added at room temperature. The mixture was then vigorously stirred at rt for 2.5 h in the dark to obtain an intensely blue fluorescing solution. The solution instantly turns blue after stirring the reaction mixture, which is clearly observed with the help of a UV lamp. The crude mixture was then purified by flash chromatography to afford a yellow solid **20** in quantitative yield (97%). Azide–alkyne click reaction G3 dendron and azide **5**, on the other hand, it afforded a yellow solid as a surface-modified G3 dendron **21** with a 99% yield (Figure 6).

The incomplete reaction or fragmentation of certain branches are very common drawbacks of a reaction involving a large number of reactions per molecule. This issue becomes serious when the size of the dendron continues to increase in convergent synthesis or during the outward growth of a dendrimer in a divergent method of synthesis. A small amount of heat during synthesis can cause the fragmentation of dendritic branches, leading to imperfections in the dendritic architecture. In our approach, these synthetic limitations were eliminated using click chemistry during the late-stage modification. This is because azide–alkyne click reaction efficiently competes at room temperature, leaving no unreacted sites in the molecule. Additionally, this reaction is compatible with any organic solvents as well as water.

### 2.6. Synthesis of G2 Homo- and Janus Dendrimers

With surface-modified polyurethane dendrons **18** and **19** in hand, we moved forward to synthesize G2 dendrimers. The synthetic protocol is illustrated in Figure 7 and Appendix A. As previously mentioned for the synthesis of G1 dendrimers, a tin catalyst (DBTDL, 8.4 eq) was used to convert an isocyanate function to a urethane linkage. To accomplish this, blue–fluorescent and non-fluorescent G2 dendrons **18** and **19** (1.2 eq each), respectively were allowed to react with hexamethylene diisocyanate core at room temperature for 20 h. Chromatographic purification of the crude mixture yielded a mixture of three dendrimers, **22–24,** as G2 PUDs. Homodendrimer **22** and Janus dendrimer **23** showed an intense fluorescence, but the homodendrimer **24** did not. The distribution of these three products was almost identical to the distribution of G1 dendrimers, with the ratio being slightly different (**22:23:24** = 1:4.3:1.3). This strategy of dendrimer synthesis could be promising from a synthetic point of view as it creates asymmetry in a dendritic molecule simply from a reaction of two different dendrons. Additionally, most of the product distribution is occupied by a heterodendrimer (Janus dendrimer). This method can be used to construct synthetically challenging JDs; otherwise, it is impossible to synthesize using the general methods shown in Figure 1.

The syntheses of G3 homo- and Janus dendrimers, as well as G3 dendrimers, are shown in Figure 8 and Appendix A, respectively. Modified G3 dendrons **20** and **21** (1.2 eq each) were reacted with 1 eq of a hexamethylene diisocyanate core in the presence of DBTDL (16.4 eq) using anhydrous DMF as a solvent to afford a mixture of G3 dendrimers **25–27**, of which **25** and **27** were homodendrimers and **26** was a Janus dendrimer. As in G2, homodendrimer **25** and Janus dendrimer **26** were blue and fluorescent, whereas the homodendrimer **27** was non-fluorescent. Nonetheless, the statistical distribution of these dendrimers was dramatically different from G1 and G2 dendrimers. The product ratio in this case was found to be 1:31:1 for **25**, **26**, and **27**, respectively. This showed that most of the product distribution was solely occupied the Janus dendrimer.

In these transformations, DBTDL acts as a catalyst. Sn coordinates with isocyanate thereby make the NCO group more susceptible to a nucleophilic attack via the focal point (OH) of dendron during dendrimer growth. However, a catalytic amount of DBTDL was not sufficient for successfully carrying out these transformations. Instead, a stoichiometric amount of this reagent was required for an excellent yield of the dendrimers because a greater portion of Sn is coordinated in a sphere created by 1,2,3-triazole. The separation of three dendrimers from the crude mixture was accomplished by gradient elution during flash chromatography, where blue–fluorescent G1–G3 homodendrimers were first eluted using acetone-DCM as a mobile phase. The polarity of the mobile phase was then gradually increased to 5% MeOH in DCM to elute blue–fluorescent G1–G3 Janus dendrimers followed by the elution of non-fluorescent homodendrimers.

### 2.7. Characterization of Dendrimers

Triazole moiety-containing G1–G3 PUDs were first characterized by homonuclear one-dimensional (1D) proton and carbon nuclear magnetic resonance (^1^H NMR and ^13^C) spectroscopy. As shown in the spectra in ESI, the peaks of similar G1–G3 homo- or Janus dendrimers (Appendix A) or G1–G3 dendrons (Appendix A) exhibited similar peak patterns. However, the peak broadening was observed when increasing the generation of dendrimers. This can be associated with the fact that the protons become slightly inequivalent with the increasing size of dendrons or dendrimers. This could partly be caused by a ‘tumbling effect.’ The slow tumbling of macromolecules in solution leads to faster relaxation of transverse magnetization due to enhanced spin–spin interactions leading to sharp peaks in the spectrum. On the other hand, a peak broadening is observed when the relaxation is faster due to an increased viscosity. The increase in the viscosity of these PUDs can be expected with increased generation due to the increased number of polar urethane linkages that can undergo intermolecular hydrogen bonding. For these reasons, peak splitting, and thus the resolution, gradually decreased in the higher-generation PU dendrons and dendrimers.

The representative ^1^H NMR spectrum of G2 Janus dendrimer **23** and G3 dendron **20** is depicted in Figure 1. The presence of two different types of NH peaks, one that is more deshielded at 9.50 ppm and the other that is slightly shielded at 6.94 ppm, demonstrate that the attachment of the focal point of the dendron (OH) to the hexamethylene diisocyanate core was successful. The more downfield peak was assigned to the NH peak in the vicinity of branching point aromatic rings, whereas the upfield peaks were assigned to the NH in the aliphatic region of the dendrimer. The most deshielded aromatic proton peak at 8.53 ppm was caused by the triazole proton, thereby providing evidence of the successful completion of an azide–alkyne click reaction. Two protons of branching aromatic ring were spotted at ~7.55 ppm and 7.20 ppm and labeled as ‘a’ and ‘b’, respectively, as shown in Figure 1.

G1–G3 dendrons and dendrimers were then investigated by high-resolution electrospray ionization–mass spectrometry (HR ESI-MS) and matrix-assisted laser desorption/ionization time-of-flight mass spectrometry (MALDI-TOF-MS). The MALDI-TOF-MS spectra of some of the dendrons and dendrimers are shown in Figure 1.

### 2.8. Photophysical Properties—Absorption and Emission

Photophysical properties of the G1–G3 dendrimers were investigated using UV–vis and fluorescence spectroscopy (Figure 2 and Table 1). As shown in Figure 2A, G1 dendrimers (**9–11**) absorbed at 343 nm and 420 nm. These absorption maxima were assigned to the dendrimers containing naphthalimide and blue fluorophore, respectively. Similarly, G2 dendrimers (**22–24**) were found to absorb at 345 nm and 417 nm due to the presence of naphthalimide and coumarin fluorophores, respectively (Figure 2C). G3 dendrimers (**25–27**) exhibited an almost similar pattern of UV absorption showing the absorption maxima at 344 nm and 418 nm (Figure 2E). In all cases, the Janus dendrimers presented two absorption maxima due to the presence of two fluorophores.

G1–G3 dendrimers did not fluoresce in solid state; however, they showed good fluorescence in a solution. A 1:1 solution of methylene chloride and methanol was chosen as the solvent to study the fluorescence of these dendrimers. The fluorescence emission spectra of the G1 dendrimers are shown in Figure 2B. Homodendrimers **9** and **11** were emitted at a maximum wavelength of 478 nm and 410, respectively, and as expected, Janus dendrimer **10** had two emission peaks at 484 nm and 414 nm. Similar observations were observed for G2 and G3 dendrimers (Figure 2B,F, respectively), where homodendrimers had one absorption peak, and the Janus dendrimers had two peaks. Moreover, the longer wavelength of each Janus dendrimer had a slight increase in its wavelength compared to the corresponding blue–fluorescent homodendrimer. For example, blue–fluorescent G3 Janus dendrimer **26** was slightly absorbed at a longer wavelength (495 nm) compared to its blue–fluorescent homodendrimer analog **25** (489 nm).

### 2.9. FRET

The bathochromic shift of the blue–fluorescent Janus dendrimer can be explained by a phenomenon called FRET, an acronym for Förster resonance energy transfer or fluorescence resonance energy transfer. For FRET to occur, the fluorescence emission of one fluorophore must overlap with the absorption of the other fluorophore. In our study, the emission caused by naphthalimide fluorophore (~350–450 nm) was found to precisely coincide the absorption of coumarin fluorophore (~350–475 nm). For this reason, the fluorescence energy from the naphthalimide fluorophore (donor) was transferred to the coumarin fluorophore (acceptor), thereby shifting the emission of Janus dendrimers toward the longer wavelength (Figure 3D). This is evident from the red-shifted emission of Janus dendrimers (Table 1). This shift of emissions toward longer wavelengths was accompanied by an increase in the intensity of emitted light. For instance, G1 Janus dendrimer **10** (shows FRET) exhibited intense fluorescence compared with homodendrimer **9** (no FRET). This red-shifting was found to be somewhat constant (6–7 nm) with the increased generation of dendrimers. This is potentially due to the fact that fluorescence resonance energy transfer is primarily affected by the distance between the donor and acceptor fluorophores, which can become closer together in a solution due to bond rotation. The FRET phenomenon was not observed when G1–G3 Janus dendrimers were irradiated, with the wavelength corresponding to the acceptor (blue) fluorophore simply because there was no possibility of energy transfer. This type of FRET measurement is useful for the detection and tracking the interactions between proteins [40], observing membrane fluidity [41], and sensing [42] applications.

Stokes shift is defined as the electronic transition between the excited and ground state of a chemical entity, which is measured as the difference between absorption and emission maxima. It is the outcome of two phenomena—vibrational relaxation and solvent reorganization. When G1–G3 dendrimers are dissolved in polar solvents such as methanol, solvent molecules surround the fluorophore. They can quickly reorientate their dipoles to stabilize the excited state of fluorophore more than the ground state. The difference in energy following solvent reorganization results in Stokes shift [43]. The G1–G3 dendrimers displayed very similar absorption (343–345 nm for naphthalimide fluorophores and 417–420 nm for blue fluorophores) and emission (413–415 nm for naphthalimide fluorophores and 478–489 nm for blue fluorophores) maxima wavelengths. This resulted in the red-shifting (∆λ) of G1–G3 homodendrimers by 68–71 nm, which is a moderately large Stokes shift. Figure 3A,C shows the Stokes shift of G2 homodendrimers.

### 2.10. Stokes shift

Unlike homodendrimers, Janus dendrimers (G1–G3) exhibited two Stokes shifts when the compound was irradiated with absorption maxima corresponding to the nathphthalimide fluorophore (343–345 nm). While the first Stokes shift was quite similar to that of homodendrimers, the second was dramatically red-shifted. Janus dendrimers of all generations exhibited large Stokes shifts (∆λ ca. 141–151 nm). As shown in Figure 3B, the G2 Janus dendrimer revealed both moderately large (4714 cm^−1^) and large (8742 cm^−1^) Stokes shifts (calculated as (1/λ_max.abs_ − 1/λ_max.em_) × 10^7^)). A large Stokes shift is an indicative of fast relaxation from the initial state to the final emission state, which is caused by intramolecular energy transfer from one part of the molecule to another part of the same molecule. In our study, naphthalimide fluorophore was found to transfer energy to the blue fluorophores. Compared with other strategies, such as excited state atom transfer [44], and by introducing alternating vibronic structures [45], this approach ensures that large Stokes shifts occur simply by installing two different fluorophores in a dendrimer in one synthetic operation. The large Stokes shift (≥100 nm) of these dendrimers is advantageous for their application in fluorescence imaging, as the wide gap between the absorption and emission causes a decrease in self-absorption.

## 3. Materials and Methods

### 3.1. General Information

Starting materials were used as obtained from commercial sources: Sigma Aldrich (NaN_3_, AIBN, dibutyltin dilaurate (DBTDL), triethylamine, hexamethylene-1,6-diisocyanate), TCI (4-pentyn-1-ol, 1-bromoundecanol), and Alfa Aesar (5-hydroxyisophthalic acid, DPPA). Whereas anhydrous solvents were used in the dendrimer synthesis, DMF (Acros Organics), DCM (Fischer Scientific), and acetone (Acros Organics) were used as received, and reagent toluene was used without distillation. Curtius reaction was set in a Carousel reactor and all other reactions were performed using classical batch process using oil bath (if heat needed). Melting points were determined using Thermo Scientific MelTemp 3.0 instrument.

^1^H, ^13^C, and 2D NMR spectra were recorded with a Bruker Advance 500 MHz NMR instrument at 298 K. NMR spectra were recorded using either acetone-*d*_6_ or CDCl_3_ as deuterated solvent and, accordingly, the solvent residual peaks were obtained at δ 2.05 ppm (qn) and δ 7.26 ppm (s), respectively, in ^1^H NMR. In ^13^C NMR, solvent residual peaks were recorded at δ 206.68 ppm (s) and δ 29.92 ppm (septet) for acetone-*d*_6_ and δ 77.23 ppm (s), respectively, for CDCl_3_. Coupling constants (J) are given in hertz (Hz), whereas chemical shifts are given in δ scale (ppm). Moreover, the multiplicities are indicated as—s (singlet), d (doublet), t (triplet), q (quartet), qn (quintet), or m (multiplet). IR spectra were obtained from PerkinElmer Spectrum One FT-IR Spectrometer.

HRMS spectra of small molecules including dendrons were obtained from ESI-LTQ-Orbitrap. MALDI of larger molecules were recorded with a Bruker Autoflex 3 instrument using ∝−cyano-4-hydroxycinnamic acid (CCA) as a matrix in positive ion mode.

Purification of compounds was carried out using flash chromatography with irregular silica of 40–60 μm, 60 Å. Small-scale purification was achieved using auto-column flash cartridges packed with 12 g or 40 g silica of 40–75 μm, 60 Å (obtained from Sorbtech and Supelco Technologies). Flow rate was 10–30 mL/min. Mobile phase used in these separations was ethyl acetate, hexane, DCM, or a mixture of these solvents.

### 3.2. Synthesis of Azide-Alkyne Clicked G1 Dendron, ***7***

4-Bromo-N-ethyl-1,8-naphthalimide, **5b** (Appendix A). Compound **5b** was synthesized by a previously reported procedure [46]. In brief, in a clean RB flask, a mixture of 4-bromo-1,8-naphthalic anhydride **5a** (2.5 g, 9.0 mmol, 1.0 eq), 70% ethylamine (3.36 mL, 36.0 mmol, 4.0 eq), and dioxane (90 mL) was refluxed for 7 h taking 2.02 mL of ethylamine as the first aliquot. After 7 h, the second aliquot of triethylamine (1.34 mL) was added and further refluxed for 14 h, cooled to room temperature, and then poured into cold water to collect a yellow solid as product **5b** (2.712 g, 99%) on filtration. This product was taken to the next step without further purification. Spectra were similar to the previously reported literature. ^1^H NMR (400 MHz, CDCl_3_): δ 8.63 (dd, *J* = 7.3, 1.2 Hz, 1H), 8.53 (dd, *J* = 8.5, 1.2 Hz, 1H), 8.38 (d, *J* = 7.9 Hz, 1H), 8.01 (d, *J* = 7.8 Hz, 1H), 7.82 (dd, *J* = 8.5, 7.3 Hz, 1H), 4.23 (q, *J* = 7.1 Hz, 2H), 1.33 (t, *J* = 7.1 Hz, 3H). ^13^C NMR (101 MHz, CDCl_3_): δ 163.41, 163.38, 133.18, 131.95, 131.14, 131.06, 130.59, 130.17, 128.95, 128.05, 123.16, 122.30, 77.36, 77.04, 76.72, 35.65, 13.32.

4-Azido-N-ethyl-1,8-naphthalimide, **5**. Compound 5 was synthesized using a previously reported procedure [36]. Briefly, in a clean RB flask, NaN_3_ (2.32 g, 35.03 mmol, 3.6 eq) in NMP (35 mL) was stirred at 50 °C for 24 hr. Then, **5b** (2.96 g, 9.73 mmol, 1.0 eq) was added at room temperature and further stirred at room temperature for 24 h. The solution was then diluted with water (100 mL) and extracted with EtOAc, washed with brine, dried over anhydrous MgSO_4_, filtered, and evaporated under reduced pressure. The crude was purified by flash chromatography using 10%–25% EtOAc in hexane to obtain a yellow solid as product **5** (2.51 g, 97%). Spectra were similar with the previously reported literature. ^1^H NMR (400 MHz, CDCl_3_): δ 8.67 (d, *J* = 7.4 Hz, 1H), 8.61 (dd, *J* = 7.9, 2.2 Hz, 1H), 8.46 (d, J = 8.4 Hz, 1H), 7.81–7.72 (m, 1H), 7.49 (dd, J = 7.9, 2.2 Hz, 1H), 4.26 (qd, *J* = 7.2, 2.0 Hz, 2H), 1.35 (td, *J* = 7.1, 2.1 Hz, 3H). ^13^C NMR (101 MHz, CDCl_3_): δ 206.95, 163.83, 163.42, 143.41, 132.17, 131.66, 129.19, 128.74, 126.86, 124.40, 122.75, 119.05, 114.68, 77.34, 77.02, 76.71, 35.53, 30.94, 13.36.

Azide-alkyne clicked G1 dendron, **7** (Appendix A). Dendron **3** (103.0 mg, 0.20 mmol, 1 eq) was dissolved in THF (2.0 mL) in an RB flask to which azidonaphthalimide **5** (160.0 mg, 0.60 mmol, 1.5 eq/triple bond) was added. After adding an aqueous solution of CuSO_4_·5H_2_O (0.15 eq/N_3_) and sodium ascorbate (0.30 eq/N_3_) in a minimum amount of water to the flask, the reaction mixture was stirred vigorously at room temperature under dark conditions. When the alkyne was completely consumed, the reaction mixture was diluted and extracted with DCM, combined organic layers were dried over anhydrous MgSO_4_, and the solvent was evaporated under reduced pressure. The crude was purified by flash chromatography using 24 % acetone in DCM as mobile phase to obtain a yellow solid as product **7** (202.1 mg, 96% yield). M.p.: 133-138 °C (no sharp melting point, just fuss to form a globular mass). TLC (24% acetone/DCM): R*_f_* 0.28. ^1^H NMR (400 MHz, DMSO-*d*_6_): δ 9.57 (s, 2H), 8.65–8.56 (m, 6H), 8.22 (d, *J* = 8.8 Hz, 2H), 8.03 (dd, *J* = 7.8, 2.1 Hz, 2H), 7.94 (t, *J* = 8.2 Hz, 2H), 7.24 (s, 1H), 6.72 (s, 2H), 4.22 (t, *J* = 6.4 Hz, 4H), 4.11 (q, *J* = 7.0 Hz, 4H), 3.81 (t, *J* = 6.5 Hz, 2H), 3.37 (s, 1H), 2.94 (t, *J* = 7.6 Hz, 4H), 2.14 (dq, *J* = 13.5, 7.5 Hz, 5H), 1.66 (t, *J* = 7.0 Hz, 2H), 1.37 (d, *J* = 9.0 Hz, 6H), 1.29–1.21 (m, 22H). ^13^C NMR (101 MHz, DMSO-*d*_6_): δ 163.32, 162.79, 159.52, 153.89, 147.48, 140.74, 138.25, 131.80, 130.80, 129.86, 129.12, 128.65, 126.08, 125.26, 124.29, 123.47, 122.90, 101.41, 99.36, 67.65, 63.91, 61.19, 35.45, 33.01, 30.06, 29.57, 29.49, 29.46, 29.43, 29.27, 29.10, 28.64, 25.97, 22.11, 13.48. HRMS (ESI-LTQ-Orbitrap) (*m/z*): [M+Na]^+^ Calcd. for C_57_H_62_N_10_O_10_Na 1069.4543; found 1069.4542. 

Experimental procedure for the synthesis of G1 dendron **6** (Appendix A has been reported previously by our group [25].

### 3.3. Synthesis of G1 Homo- and Heterodendrimers, ***9**–**11***

An oven-dried RB flask equipped with a magnetic stir bar was charged with dendrons **6** (86.7 mg, 0.084 mmol, 1.2 eq) and **7** (88.0 mg, 0.084 mmol, 1.2 eq). After flushing and backfilling with N_2_, dry DMF was added to the flask via syringe. Then hexamethylene -1,6-diisocyanate **8** (11.2 μL, 0.070 mmol, 1.0 eq) and dibutyltin dilaurate (DBTDL) (219 μL, 0.370 mmol, 4.4 eq) were added successively. The reaction mixture was stirred vigorously at room temperature for 20 h before diluting and extracting with DCM. The combined organic layers were washed multiple times with saturated NaHCO_3_ and water to remove excess DBTDL and DMF, washed with brine, dried over anhydrous MgSO_4_, filtered, and evaporated under educed pressure. The crude was then purified with flash chromatography using 24% acetone in DCM as mobile phase to obtain a yellow solid as product (135.9 mg, 91% overall yield).

Homodendrimer, **9** (14.5 mg) (characterization reported previously) [25]. 

Janus dendrimer, **10** (104.1 mg). M.p.: 118–128 °C; TLC (25% acetone/DCM): R*_f_* 0.24; ^1^H NMR (500 MHz, CDCl_3_): δ 8.65 (tt, *J* = 7.9, 5.6 Hz, 4H), 8.40–8.22 (m, 6H), 7.96–7.87 (m, 2H), 7.84–7.75 (m, 4H), 7.40 (dd, *J* = 8.9, 5.7 Hz, 2H), 7.25 (s, 4H), 7.12 (d, *J* = 5.8 Hz, 2H), 7.04 (s, 1H), 6.88–6.67 (m, 6H), 6.59 (d, *J* = 7.8 Hz, 2H), 4.82 (s, 1H), 4.27 (dtd, *J* = 17.7, 6.2, 2.5 Hz, 12H), 4.04 (t, *J* = 6.7 Hz, 4H), 3.93–3.79 (m, 4H), 3.47 (q, *J* = 7.1 Hz, 8H), 3.16 (d, *J* = 7.5 Hz, 4H), 3.01 (t, *J* = 7.2 Hz, 4H), 2.93 (dt, *J* = 7.6, 3.7 Hz, 5H), 2.13 (s, 4H), 1.70 (d, *J* = 7.2 Hz, 4H), 1.63–1.54 (m, 4H), 1.52–1.46 (m, 4H), 1.40–1.22 (m, 55H). ^13^C NMR (126 MHz, CDCl_3_): δ 163.48, 163.45, 162.95, 162.93, 160.26, 156.93, 155.61, 153.49, 151.02, 147.66, 146.75, 139.54, 138.23, 134.63, 132.06, 132.03, 130.62, 130.60, 130.07, 129.57, 129.54, 128.99, 128.49, 128.47, 126.32, 126.30, 126.28, 123.67, 123.37, 122.93, 122.27, 117.28, 110.62, 99.88, 97.69, 77.31, 77.05, 76.80, 69.53, 68.05, 68.00, 64.92, 64.38, 64.11, 63.00, 53.81, 45.46, 40.73, 35.77, 32.78, 31.76, 30.96, 29.92, 29.71, 29.46, 29.42, 29.39, 29.36, 29.32, 29.28, 29.20, 29.15, 29.05, 28.51, 28.44, 26.26, 25.96, 25.94, 25.86, 25.83, 25.72, 22.33, 22.07, 13.32, 12.34. HRMS (ESI-LTQ-Orbitrap) (*m/z*): [M+Na]^+^ Calcd for C_120_H_145_N_22_O_22_Na 2269.0748; found 2269.0743.

Homodendrimer, **11** (17.3 mg); M.p.: 118–137 °C; TLC (25% acetone/DCM): R*_f_* 0.10; ^1^H NMR (500 MHz, DMSO-*d*_6_): δ 9.56 (s, 4H), 8.59 (s, 4H), 8.56–8.48 (m, 8H), 8.18 (dd, *J* = 8.6, 1.1 Hz, 4H), 7.99 (d, *J* = 7.8 Hz, 4H), 7.88 (dd, *J* = 8.6, 7.2 Hz, 4H), 7.23 (t, *J* = 1.8 Hz, 2H), 6.99 (t, *J* = 5.8 Hz, 2H), 6.72 (d, *J* = 1.9 Hz, 4H), 4.22 (t, *J* = 6.4 Hz, 8H), 4.07 (q, *J* = 7.1 Hz, 8H), 3.88 (t, *J* = 6.6 Hz, 4H), 3.79 (t, *J* = 6.5 Hz, 4H), 2.92 (dt, *J* = 16.4, 7.1 Hz, 12H), 2.13 (t, *J* = 7.1 Hz, 8H), 1.64 (t, *J* = 7.4 Hz, 4H), 1.53–1.43 (m, 4H), 1.36–1.18 (m, 50H); ^13^C NMR (126 MHz, DMSO*-d*_6_): δ 163.32, 162.79, 159.52, 156.77, 153.88, 147.47, 140.74, 138.25, 131.80, 130.79, 129.86, 129.11, 128.65, 126.07, 125.24, 124.27, 123.47, 122.90, 101.37, 99.34, 68.97, 67.64, 63.93, 63.90, 56.30, 55.39, 35.45, 32.58, 31.16, 30.07, 29.84, 29.45, 29.40, 29.26, 29.19, 29.10, 28.63, 26.37, 25.97, 25.86, 22.11, 13.48; MALDI-TOF-MS (Ferulic acid matrix) (*m/z*): [M+H]^+^ Calcd. for C_122_H_137_N_22_O_22_ 2262.02; found 2263.11.

### 3.4. Growth of Dendrons, ***12**–**16***

G2 phenolic dendron, **13**. An oven-dried carousel tube was charged with 5-hydroxy-1,3-dicarbonyl diazide **12** (431.2 mg, 1.86 mmol, 1.0 eq) and G1 dendron 3 (1.35 g, 2.6 mmol, 1.5 eq). After degassing and backfilling the tube with nitrogen, anhydrous DMF was added, and then the solution was transferred to the carousel reactor maintained at 95 °C. The reaction was monitored with TLC. After 72 h, the reaction mixture was diluted with EtOAc, extracted, combined organic layers were washed multiple times with water then washed with brine once, dried over anhydrous MgSO_4_, filtered, and evaporated under reduced pressure. The crude was purified by flash chromatography using 40% EtOAc in hexane to obtain a colorless solid as product **13** (531.6 mg, 65%). M.p. 70 °C; TLC (40% EtOAC in hexane): R*_f_* 0.25; **^1^**H NMR (500 MHz, CD_3_COCD_3_): δ 8.63 (s, 4H), 8.51 (s, 2H), 7.28 (t, *J* = 1.9 Hz, 2H), 7.20 (t, *J* = 1.9 Hz, 1H), 6.98 (d, *J* = 2.0 Hz, 4H), 6.90 (d, *J* = 1.9 Hz, 2H), 4.21 (t, *J* = 6.3 Hz, 8H), 4.10 (t, *J* = 6.6 Hz, 4H), 3.95 (t, *J* = 6.5 Hz, 4H), 2.40 (t, *J* = 2.7 Hz, 4H), 2.33 (td, *J* = 7.1, 2.7 Hz, 8H), 1.87 (q, *J* = 6.7 Hz, 8H), 1.80–1.73 (m, 4H), 1.65 (p, *J* = 6.7 Hz, 4H), 1.48 (dq, *J* = 12.1, 6.8 Hz, 4H), 1.42–1.33 (m, 24H); ^13^C NMR (126 MHz, CD_3_COCD_3_): δ 160.10, 158.11, 153.53, 153.37, 140.69, 140.58, 100.71, 100.01, 99.89, 99.19, 82.99, 69.56, 69.53, 67.56, 64.28, 62.93, 29.51, 28.95, 27.95, 25.88, 25.72, 14.54; HRMS (ESI-LTQ-Orbitrap) (*m/z*): [M+Na]^+^ Calcd. for C_66_H_88_N_6_O_15_Na 1227.6200; found 1227.6205.

G2 dendron, **14**. An oven-dried 50 mL RB flask was charged with 11-bromoundecanol (340.6 mg, 1.36 mmol, 1.5 eq), K_2_CO_3_ (624.3 mg, 4.52 mmol, 5.0 eq), KI (30.0 mg, 0.18 mmol, 0.2 eq), anhydrous acetone (10 mL), and a magnetic stir bar. After degassing and filling the flask with nitrogen, **13** (1089.6 mg, 0.90 mmol, 1.0 eq) in 5 mL anhydrous acetone was transferred into the flask via syringe. The reaction mixture was set to reflux under nitrogen. Progress of the reaction was monitored with TLC (3:2 hexane/EtOAc). After 20 h, solvent was evaporated, crude was extracted with EtOAc, washed with brine, dried with anhydrous MgSO_4_, concentrated, and finally purified by flash chromatography using 40% EtOAc in hexane as mobile phase to give a slightly yellow viscous oil as product **14** (1112.0 mg, 90% yield). TLC (40% EtOAc in hexane): R*_f_* 0.19; ^1^H NMR (500 MHz, CD_3_COCD_3_): δ 8.72–8.49 (m, 6H), 7.28 (dt, *J* = 4.6, 1.8 Hz, 3H), 6.99 (d, *J* = 1.9 Hz, 6H), 4.21 (t, *J* = 6.3 Hz, 8H), 4.12 (t, *J* = 6.6 Hz, 4H), 3.95 (t, *J* = 6.5 Hz, 6H), 3.55 (td, *J* = 6.6, 5.3 Hz, 2H), 3.40 (t, *J* = 5.3 Hz, 1H), 2.40 (t, *J* = 2.7 Hz, 3H), 2.32 (td, *J* = 7.2, 2.7 Hz, 9H), 1.90–1.84 (m, 8H), 1.81–1.73 (m, 6H), 1.70–1.62 (m, 4H), 1.54–1.44 (m, 8H), 1.42–1.31 (m, 38H); ^13^C NMR (126 MHz, CD_3_COCD_3_): δ 160.09, 160.06, 153.63, 153.42, 140.66, 140.57, 100.77, 99.22, 99.16, 83.02, 69.56, 67.58, 67.57, 64.41, 62.95, 61.63, 32.90, 29.54, 29.53, 28.99, 27.95, 25.90, 25.89, 25.81, 25.73, 14.55; HRMS (ESI-LTQ-Orbitrap) (*m/z*): [M+Na]^+^ Calcd for C_77_H_110_N_6_O_16_Na 1397.7871; found 1397.7973.

G3 phenolic dendron, **15**. Similar procedure as that of **13**, synthesis was employed using **12** (119.2 mg, 0.513 mmol, 1.0 eq), **14** (1057.6 mg, 0.770 mmol, 1.5 eq), and anhydrous DMF (6 mL) to afford a highly viscous, transparent oil as product **15** (596.4 mg, 62%) after flash chromatography (30-50% EtOAc in hexane with the compound eluting in 50 % EtOAc, unreacted **14** elutes in 40% EtOAc). TLC (50% EtOAc in hexane): R*_f_* 0.34; ^1^H NMR (500 MHz, CD_3_COCD_3_): δ 8.67–8.46 (m, 11H), 7.28 (dt, *J* = 3.6, 1.9 Hz, 6H), 7.20 (t, *J* = 1.9 Hz, 1H), 6.99 (d, *J* = 2.0 Hz, 12H), 6.90 (d, *J* = 1.9 Hz, 2H), 4.21 (t, *J* = 6.3 Hz, 16H), 4.15–4.09 (m, 12H), 3.95 (t, *J* = 6.6 Hz, 12H), 2.39 (t, *J* = 2.7 Hz, 7H), 2.32 (td, *J* = 7.1, 2.7 Hz, 17H), 1.91–1.85 (m, 16H), 1.80–1.74 (m, 12H), 1.69–1.62 (m, 12H), 1.47 (q, *J* = 7.3 Hz, 12H), 1.42–1.31 (m, 80H); ^13^C NMR (126 MHz, CD_3_COCD_3_): δ 170.01, 160.10, 160.07, 153.59, 153.54, 153.38, 153.31, 140.67, 140.58, 140.48, 100.72, 100.02, 99.19, 82.99, 69.53, 67.57, 67.55, 64.39, 64.30, 62.93, 59.64, 29.51, 29.51, 28.95, 27.95, 25.89, 25.73, 19.93, 14.55, 13.61; MALDI-TOF-MS (CCA matrix) (*m/z*): [M+Na]^+^ Calcd. for C_162_H_224_N_14_O_35_Na 2948.61; found 2951.01

G3 dendron, **16**. Similar procedure as that of **14,** synthesis was employed using **15** (217.2 mg, 0.0742 mmol, 1.0 eq), 11-bromoundecanol (28.0 mg, 0.113 mmol, 1.5 eq), K_2_CO_3_ (51.3 mg, 0.371 mmol, 5.0 eq), KI (2.5 mg, 0.0150 mmol, 0.2 eq), and anhydrous acetone (3 mL) to obtain a highly viscous, transparent oil as product 16 (184.0 mg, 80%) after flash chromatography (1:1 hexane/EtOAc). TLC (50% EtOAc in hexane): R*_f_* 0.32; ^1^H NMR (500 MHz, CD_3_COCD_3_): δ 8.67–8.54 (m, 12H), 7.28 (q, *J* = 1.9 Hz, 7H), 6.99 (d, *J* = 2.0 Hz, 14H), 4.21 (t, *J* = 6.3 Hz, 16H), 4.11 (t, *J* = 6.6 Hz, 12H), 3.95 (td, *J* = 6.5, 1.7 Hz, 14H), 2.40 (t, *J* = 2.7 Hz, 7H), 2.32 (td, *J* = 7.1, 2.7 Hz, 17H), 1.90–1.84 (m, 16H), 1.77 (qd, *J* = 6.6, 4.5 Hz, 14H), 1.66 (p, *J* = 6.6 Hz, 12H), 1.53–1.45 (m, 16H), 1.42–1.31 (m, 92H); ^13^C NMR (126 MHz, CD_3_COCD_3_): δ 170.01, 160.10, 160.07, 153.59, 153.53, 153.38, 153.31, 140.67, 140.58, 140.48, 100.72, 99.19, 82.99, 69.54, 67.57, 67.55, 64.39, 62.93, 61.61, 59.65, 32.92, 27.96, 25.89, 25.87, 25.82, 25.74, 19.94, 14.55, 13.61; MALDI-TOF-MS (FA matrix) (*m/z*): [M+Na]^+^ Calcd. for C_162_H_224_N_14_O_35_Na 3118.77; found 3120.47

### 3.5. Late-Stage Modification of G2 and G3 Dendrons 

General procedure of copper-catalyzed azide-alkyne cycloaddition (CuAAC). 

Compound with terminal alkyne (1 eq) was dissolved in THF in an RB flask to which azidocoumarin (1.2 eq/triple bond) was added. After adding an aqueous solution of CuSO_4_·5H_2_O (0.15 eq/N_3_) and sodium ascorbate (0.30 eq/N_3_) in minimum amount of water to the flask, the reaction mixture was stirred vigorously at room temperature under dark conditions. When the alkyne was completely consumed, the reaction mixture was diluted and extracted with DCM, combined organic layers were dried over anhydrous MgSO_4_, and the solvent was evaporated under reduced pressure. The crude was purified by flash chromatography. Alternatively, the crude can be precipitated in water to obtain the solid product. However, this could also precipitate excess azido compounds as impurities of the product. 

Blue fluorescent G2 dendron **18**. General procedure of CuAAC was followed using dendron **14** (125.4 mg, 0.160 mmol, 1.0 eq), azidocoumarin **4** (198.0 mg, 0.768 mmol, 4.8 eq), CuSO_4_·5H_2_O (15.3 mg, 0.096 mmol, 0.6 eq), sodium ascorbate (38.0 mg, 0.192 mmol, 1.2 eq), THF (2 mL), and water (200 μL) to obtain a blue fluorescing yellow solid **18** (367.9 mg, 96%) after flash chromatography (90:9:1 EtOAC/Hexane/Et_3_N). M.p.: 110–115 °C; TLC (90:9:1 EtOAc/Hexane/Et_3_N): R*_f_* 0.12; ^1^H NMR (500 MHz, DMSO-*d*_6_): δ 9.64–9.44 (m, 6H), 8.44–8.28 (m, 8H), 7.60 (d, *J* = 9.0 Hz, 4H), 7.23–7.17 (m, 3H), 6.85–6.71 (m, 10H), 6.64 (d, *J* = 2.4 Hz, 4H), 4.14 (t, *J* = 6.4 Hz, 8H), 4.03 (t, *J* = 6.6 Hz, 4H), 3.83 (td, *J* = 6.7, 3.1 Hz, 6H), 3.46 (q, *J* = 7.0 Hz, 16H), 3.37 (d, *J* = 6.5 Hz, 2H), 2.83 (t, *J* = 7.6 Hz, 8H), 2.02 (p, *J* = 6.6 Hz, 8H), 1.68–1.55 (m, 10H), 1.39–1.22 (m, 48H), 1.14 (t, *J* = 7.0 Hz, 24H); ^13^C NMR (126 MHz, DMSO-*d*_6_): δ 159.59, 159.54, 157.21, 156.03, 153.95, 153.88, 151.81, 146.55, 140.82, 140.79, 136.98, 130.91, 123.33, 116.84, 110.40, 106.97, 101.45, 99.41, 96.81, 73.49, 72.68, 70.27, 67.67, 64.52, 63.89, 61.19, 55.78, 46.17, 44.67, 33.03, 31.16, 30.54, 30.07, 29.58, 29.50, 29.48, 29.44, 29.28, 29.20, 29.10, 29.00, 28.77, 25.98, 25.87, 21.98, 12.75; MALDI-TOF-MS (Ferulic acid matrix) (*m/z*): [M+H]^+^ Calcd. for C_129_H_167_N_22_O_24_ 2408.25; found 2409.79.

G2 Dendron, **19**. General procedure of CuAAC was followed using dendron **14** (110.0 mg, 0.080 mmol, 1.0 eq), azidocoumarin **17** (102.2 mg, 0.384 mmol, 4.8 eq), CuSO_4_·5H_2_O (7.7 mg, 0.048 mmol, 0.6 eq), sodium ascorbate (19.0 mg, 0.096 mmol, 1.2 eq), THF (1.5 mL), and water (200 μL) to obtain a yellow solid **19** (144.6 mg, 87%) after flash chromatography (25% acetone/DCM). M.p.: 119–133 °C (no sharp melting point); TLC (28% acetone in DCM): R*_f_* 0.30; ^1^H NMR (500 MHz, DMSO-*d*_6_): δ 9.62–9.42 (m, 6H), 8.61–8.48 (m, 13H), 8.20 (dd, *J* = 8.6, 1.1 Hz, 4H), 8.00 (d, *J* = 7.8 Hz, 4H), 7.91 (dd, *J* = 8.6, 7.3 Hz, 4H), 7.24–7.16 (m, 3H), 6.75–6.67 (m, 6H), 4.22 (t, *J* = 6.3 Hz, 8H), 4.09 (q, *J* = 7.0 Hz, 8H), 4.02 (t, *J* = 6.6 Hz, 4H), 3.84–3.75 (m, 6H), 3.38–3.35 (m, 2H), 2.94 (t, *J* = 7.5 Hz, 8H), 2.16–2.10 (m, 8H), 1.67–1.55 (m, 10H), 1.24 (t, *J* = 7.0 Hz, 62H); ^13^C NMR (126 MHz, DMSO-*d*_6_): δ 163.37, 162.84, 159.51, 153.92, 153.88, 147.47, 140.78, 140.74, 138.28, 131.84, 130.82, 129.88, 129.15, 128.70, 126.13, 125.28, 124.33, 123.54, 122.97, 101.41, 99.36, 67.64, 64.50, 63.90, 61.18, 35.46, 33.01, 31.17, 30.07, 29.57, 29.48, 29.44, 29.40, 29.25, 29.23, 29.17, 29.07, 28.99, 28.63, 25.97, 25.95, 25.85, 22.11, 13.50; MALDI-TOF-MS (Ferulic acid matrix) (*m/z*): [M+Na]^+^ Calcd. for C_133_H_150_N_22_O_24_Na 2462.11; found 2464.39

Blue fluorescent G3 dendron, **20**. General procedure of CuAAC was followed using dendron **16** (124.8 mg, 0.0403 mmol, 1.0 eq), azidocoumarin **4** (100.0 mg, 0.3870 mmol, 9.6 eq), CuSO_4_·5H_2_O (7.7 mg, 0.0484 mmol, 1.2 eq), sodium ascorbate (19.2 mg, 0.0967 mmol, 2.4 eq), THF (1.5 mL), and water (200 μL) to obtain a blue fluorescing yellow solid **20** (202.4 mg, 97%) after flash chromatography (4% methanol in DCM). M.p.: 116–127 °C; TLC (5% MeOH in DCM): R*_f_* 0.42; ^1^H NMR (500 MHz, DMSO-*d*_6_): δ 9.62–9.45 (m, 13H), 8.42–8.28 (m, 16H), 7.59 (d, *J* = 8.9 Hz, 8H), 7.19 (d, *J* = 10.0 Hz, 7H), 6.82–6.71 (m, 23H), 6.63 (d, *J* = 2.4 Hz, 9H), 4.14 (t, *J* = 6.4 Hz, 16H), 4.02 (t, *J* = 6.5 Hz, 12H), 3.82 (d, *J* = 5.9 Hz, 16H), 3.46 (q, *J* = 7.0 Hz, 32H), 2.83 (t, *J* = 7.6 Hz, 16H), 2.51 (t, *J* = 1.9 Hz, 16H), 2.02 (p, *J* = 6.8 Hz, 16H), 1.68–1.52 (m, 28H), 1.23 (q, *J* = 6.1 Hz, 126H), 1.13 (t, *J* = 7.0 Hz, 48H); ^3^C NMR (126 MHz, DMSO-*d*_6_): δ 159.59, 159.54, 157.19, 156.02, 153.93, 153.87, 151.79, 146.54, 140.81, 140.79, 136.93, 130.90, 123.30, 116.82, 110.39, 106.96, 101.44, 99.39, 96.79, 67.66, 64.50, 63.88, 63.02, 61.18, 52.47, 46.10, 44.66, 33.03, 31.77, 29.59, 29.51, 29.48, 29.46, 29.42, 29.28, 29.25, 29.21, 29.18, 29.11, 29.08, 29.00, 28.77, 25.98, 25.96, 25.87, 22.57, 21.98, 14.41, 12.74, 11.52, 7.65; MALDI-TOF-MS (BHB matrix) (*m/z*): [M+H]^+^ Calcd. for C_277_H_359_N_46_O_52_ 5161.69; found 5159.41.

G3 dendron, **21**. General procedure of CuAAC was followed using dendron **16** (80.0 mg, 0.026 mmol, 1.0 eq), azidocoumarin **17** (83.1 mg, 0.312 mmol, 9.6 eq), CuSO_4_·5H_2_O (5.0 mg, 0.0312 mmol, 1.2 eq), sodium ascorbate (12.4 mg, 0.0624 mmol, 2.4 eq), THF (1 mL), and water (150 μL) to obtain a yellow solid **21** (134.1 mg, 99%) after flash chromatography (5% acetone/DCM). M.p.: 143–152 °C (no sharp melting point); TLC (5% acetone in DCM): R*_f_* 0.11; ^1^H NMR (500 MHz, DMSO-*d*_6_): δ 9.61–9.37 (m, 14H), 8.60–8.43 (m, 28H), 8.18 (dd, *J* = 8.5, 1.1 Hz, 9H), 8.01–7.93 (m, 9H), 7.88 (dd, *J* = 8.6, 7.2 Hz, 9H), 7.24–7.12 (m, 7H), 6.71 (d, *J* = 4.7 Hz, 14H), 4.20 (d, *J* = 6.5 Hz, 18H), 4.10–3.97 (m, 32H), 3.77 (d, *J* = 6.7 Hz, 14H), 2.93 (t, *J* = 7.5 Hz, 18H), 2.12 (d, *J* = 6.7 Hz, 16H), 1.69–1.49 (m, 30H), 1.36–1.13 (m, 150H); ^13^C NMR (126 MHz, DMSO-*d*_6_): δ 206.97, 163.31, 162.78, 159.51, 153.87, 147.46, 140.73, 138.24, 131.79, 130.78, 129.84, 129.09, 128.65, 126.06, 125.22, 124.24, 123.46, 122.90, 101.36, 99.32, 67.61, 64.48, 63.89, 61.18, 55.39, 35.44, 33.01, 31.17, 29.57, 29.44, 29.39, 29.23, 29.17, 29.07, 28.98, 28.62, 25.95, 25.84, 22.10, 13.47; MALDI-TOF-MS (FA matrix) (*m/z*): [M+H]^+^ Calcd. for C_285_H_327_N_46_O_52_ 2948.61; found 2951.01

### 3.6. Synthesis of G2 Homo- and Heterodendrimers (***22**–**24***)

General procedure of the attachment of dendron/s to the core.

An oven-dried RB flask equipped with a magnetic stir bar was charged with dendron/s (fluorescent or non-fluorescent). After flushing and backfilling with N_2_, dry DMF was added to the flask via syringe. Then, hexamethylene-1,6-diisocyanate and dibutyltin dilaurate (DBTDL) were added successively. The reaction mixture was stirred vigorously at room temperature for 20 h before diluting and extracting with DCM. The combined organic layers were washed multiple times with water to remove DMF, washed with brine, dried over anhydrous MgSO_4_, filtered, and evaporated under reduced pressure. The crude was then purified with flash chromatography.

G2 dendrimers **22–24**. General method of dendron attachment was performed using dendron **18** (69.0 mg, 0.0287 mmol, 1.2 eq), dendron **19** (70.0 mg, 0.0287 mmol, 1.2 eq), hexamethylene diisocyanate **8** (4.0 μL, 0.0239 mmol, 1.0 eq), DBTDL (143 μL, 0.2411 mmol, 8.4 eq), and DMF (600 μL) to obtain a mixture of three yellow solids as products **22–24** (108.5 mg, 90% overall) after flash chromatography (gradient elution, 20–30 % acetone in DCM followed by 5% MeOH in DCM).

Homodendrimer **22** (16.8 mg); M.p. 116–125 °C; TLC (30% acetone in DCM): R*_f_* 0.50; ^1^H NMR (500 MHz, DMSO-*d*_6_): δ 9.65–9.44 (m, 12H), 9.35 (s, 6H), 8.43–8.28 (m, 13H), 7.96 (s, 2H), 7.63–7.55 (m, 6H), 7.24–7.15 (m, 6H), 7.08 (d, *J* = 7.4 Hz, 2H), 6.86–6.72 (m, 18H), 6.67–6.59 (m, 6H), 4.13 (d, *J* = 6.6 Hz, 14H), 4.05–3.96 (m, 10H), 3.82 (s, 12H), 3.53–3.42 (m, 26H), 3.09 (qd, *J* = 7.3, 4.4 Hz, 46H), 2.83 (t, *J* = 7.4 Hz, 15H), 2.02 (d, *J* = 10.2 Hz, 14H), 1.70–1.51 (m, 22H), 1.33–1.12 (m, 202H); ^13^C NMR (126 MHz, DMSO-*d*_6_): δ 174.96, 162.78, 159.58, 159.54, 157.23, 156.03, 153.95, 153.88, 151.81, 146.55, 140.82, 140.79, 137.04, 130.92, 127.88, 123.36, 116.83, 114.32, 110.42, 106.95, 101.44, 99.40, 96.80, 67.67, 64.51, 63.89, 61.18, 56.29, 54.38, 52.47, 46.09, 44.67, 36.25, 34.77, 34.67, 34.13, 33.75, 33.02, 32.58, 31.76, 31.23, 30.06, 29.58, 29.48, 29.44, 29.37, 29.27, 29.21, 29.18, 29.10, 29.00, 28.76, 25.97, 25.86, 25.62, 24.96, 23.09, 22.66, 22.57, 21.97, 21.53, 15.54, 15.22, 14.42, 12.99, 12.76, 12.74, 11.64, 9.95, 9.02, 7.65; MALDI-TOF-MS (Ferulic acid matrix) (*m/z*): [M+H]^+^ Calcd. for C_266_H_345_N_46_O_50_ 4983.58; found 4982.63.

Janus dendrimer **23** (70.2 mg); M.p. 141–150 °C; TLC (30% acetone in DCM): R*_f_* 0.33

^1^H NMR (500 MHz, DMSO-*d*_6_): δ 9.64–9.31 (m, 10H), 8.58–8.22 (m, 18H), 8.16 (d, *J* = 8.5 Hz, 5H), 8.00–7.73 (m, 9H), 7.55 (dd, *J* = 9.2, 4.0 Hz, 3H), 7.28–7.09 (m, 6H), 6.94 (s, 2H), 6.81–6.66 (m, 11H), 6.63–6.51 (m, 3H), 4.26–4.10 (m, 14H), 4.09–3.92 (m, 18H), 3.76 (d, *J* = 8.0 Hz, 12H), 3.48–3.37 (m, 8H), 2.99–2.77 (m, 14H), 2.12 (d, *J* = 8.0 Hz, 8H), 2.01 (t, *J* = 7.0 Hz, 5H), 1.73–1.46 (m, 22H), 1.39–1.08 (m, 108H); ^13^C NMR (126 MHz, DMSO-*d*_6_): δ 206.94, 163.25, 162.72, 159.58, 159.51, 157.14, 155.96, 153.86, 151.74, 147.44, 146.54, 140.72, 138.19, 136.81, 131.72, 130.84, 130.74, 129.81, 129.01, 128.60, 125.98, 125.15, 124.15, 123.37, 122.81, 116.79, 110.33, 106.93, 101.34, 99.31, 96.74, 67.61, 64.48, 63.88, 61.18, 48.11, 44.64, 35.42, 33.01, 31.15, 29.58, 29.49, 29.45, 29.41, 29.25, 29.19, 29.09, 28.99, 28.77, 28.63, 26.38, 25.96, 25.85, 24.21, 22.09, 21.98, 13.44, 12.73; MALDI-TOF-MS (Ferulic acid matrix) (*m/z*): [M+H]^+^ Calcd. for C_270_H_328_N_46_O_50_ 5017.87; found 5016.65

Homodendrimer **24** (21.5 mg); M.p. 121–130 °C; TLC (30% acetone in DCM): R*_f_* 0.17; ^1^H NMR (500 MHz, DMSO-*d*_6_): δ 9.60–9.39 (m, 12H), 8.60–8.44 (m, 27H), 8.21–8.13 (m, 9H), 7.98 (dd, *J* = 11.5, 7.7 Hz, 9H), 7.91–7.82 (m, 9H), 7.24–7.14 (m, 6H), 6.97 (d, *J* = 5.6 Hz, 2H), 6.71 (d, *J* = 4.4 Hz, 12H), 4.27–4.16 (m, 18H), 4.09–3.98 (m, 26H), 3.86 (t, *J* = 6.7 Hz, 4H), 3.79 (q, *J* = 6.3 Hz, 12H), 2.92 (q, *J* = 8.4 Hz, 22H), 2.15–2.10 (m, 16H), 1.66–1.42 (m, 44H), 1.32–1.18 (m, 232H); ^3^C NMR (126 MHz, DMSO-*d*_6_): δ 206.93, 163.33, 163.28, 162.80, 162.75, 159.51, 156.75, 153.86, 147.45, 140.73, 138.26, 138.23, 131.77, 130.77, 129.85, 129.10, 129.06, 128.67, 128.63, 126.08, 126.02, 125.24, 125.19, 124.26, 124.21, 123.48, 123.41, 122.91, 122.85, 101.35, 99.32, 67.61, 64.48, 63.89, 35.43, 34.19, 31.77, 31.63, 31.15, 29.84, 29.47, 29.40, 29.36, 29.25, 29.23, 29.18, 29.13, 29.09, 29.04, 29.00, 28.89, 28.63, 27.31, 26.37, 26.21, 25.96, 25.85, 25.07, 22.57, 22.51, 22.10, 14.41, 14.05, 13.46; MALDI-TOF-MS (Ferulic acid matrix) (*m/z*): [M+H]^+^ Calcd. for C_274_H_313_N_46_O_50_ 5047.34; found 5046.25

### 3.7. Synthesis of G3 Homo- and Heterodendrimers ***25**–**27***


General method of dendron attachment was performed using dendron **20** (85.0 mg, 0.0165 mmol, 1.2 eq), dendron **21** (86.3 mg, 0.0165 mmol, 1.2 eq), hexamethylene diisocyanate **8** (2.5 μL, 0.015 mmol, 1.0 eq), DBTDL (177 μL, 0.2805 mmol, 16.4 eq), and DMF (600 μL) to obtain a mixture of three yellow solids as products **22–24** (142.9 mg, 89% overall) after flash chromatography (gradient elution, 20–30 % acetone in DCM followed by 5% MeOH in DCM).

Homodendrimer, **25** (4.5 mg); M.p. 115–122 °C; TLC (5% MeOH in DCM): R*_f_* 0.30; ^1^H NMR (500 MHz, CDCl_3_): δ 8.29 (d, *J* = 11.0 Hz, 33H), 7.36 (d, *J* = 8.9 Hz, 30H), 7.25 (d, *J* = 16.2 Hz, 17H), 7.04 (d, *J* = 8.5 Hz, 14H), 6.84 (s, 27H), 6.64 (dd, *J* = 9.0, 2.4 Hz, 17H), 6.51 (d, *J* = 2.4 Hz, 17H), 4.23 (t, *J* = 6.1 Hz, 34H), 4.10 (t, *J* = 6.7 Hz, 26H), 3.88 (t, *J* = 6.4 Hz, 30H), 3.42 (q, *J* = 7.1 Hz, 66H), 2.89 (t, *J* = 7.5 Hz, 32H), 2.10 (t, *J* = 7.0 Hz, 34H), 1.76–1.52 (m, 62H), 1.42–1.17 (m, 341H), 1.04 (t, *J* = 7.2 Hz, 64H); ^13^C NMR (126 MHz, CDCl_3_): δ 160.23, 159.75, 157.13, 155.72, 153.80, 153.59, 151.47, 146.87, 139.68, 139.60, 134.77, 129.98, 122.14, 116.97, 110.01, 107.09, 101.01, 99.88, 96.92, 68.28, 67.99, 67.96, 65.31, 64.41, 62.85, 53.47, 52.24, 46.16, 44.96, 32.78, 29.70, 29.53, 29.47, 29.43, 29.41, 29.38, 29.36, 29.29, 29.26, 29.22, 29.18, 29.14, 28.90, 28.47, 25.94, 25.91, 25.82, 25.76, 22.38, 12.43, 11.41; MALDI-TOF-MS (DHB matrix) (*m/z*): [M+H]^+^ Calcd. for C_562_H_728_N_94_O_106_ 10494.56; molecular ion was not detected with any matrix. This may be due the fact that the molecule absorbed close to the wavelength of laser, thereby fragmentating the parent ion before it was detected [47]. 

Janus dendrimer, **26** (133.2 mg); M.p. 146–154 °C; TLC (5% MeOH in DCM): R*_f_* 0.28; ^1^H NMR (500 MHz, DMSO-*d*_6_): δ 9.62–9.36 (m, 12H), 8.59–8.40 (m, 17H), 8.39–8.25 (m, 5H), 8.15 (t, *J* = 6.2 Hz, 6H), 7.94 (t, *J* = 7.3 Hz, 6H), 7.88–7.74 (m, 6H), 7.54 (q, *J* = 9.1 Hz, 3H), 7.23–7.10 (m, 7H), 6.95 (s, 1H), 6.80–6.66 (m, 15H), 6.64–6.51 (m, 3H), 4.25–4.10 (m, 18H), 4.09–3.94 (m, 24H), 3.90–3.67 (m, 16H), 3.43 (t, *J* = 7.5 Hz, 10H), 2.97–2.79 (m, 18H), 2.12 (d, *J* = 7.1 Hz, 8H), 2.01 (d, *J* = 9.4 Hz, 6H), 1.71–1.48 (m, 28H), 1.34–1.09 (m, 144H); ^13^C NMR (126 MHz, DMSO-*d*_6_): δ 206.94, 163.25, 162.72, 159.58, 159.51, 157.14, 155.96, 153.86, 151.74, 147.44, 146.54, 140.72, 138.19, 136.81, 131.72, 130.84, 130.74, 129.81, 129.01, 128.60, 125.98, 125.15, 124.15, 123.37, 122.81, 116.79, 110.33, 106.93, 101.34, 99.31, 96.74, 67.61, 64.48, 63.88, 61.18, 48.11, 44.64, 35.42, 33.01, 31.15, 29.58, 29.49, 29.45, 29.41, 29.25, 29.19, 29.09, 28.99, 28.77, 28.63, 26.38, 25.96, 25.85, 24.21, 22.09, 21.98, 13.44, 12.73; MALDI-TOF-MS (DHB matrix) (*m/z*): [M+H]^+^ Calcd. for C_562_H_729_N_94_O_106_ 10560.39; found 10478.22

Homodendrimer, **27** (5.2 mg); M.p. 121–130 °C; TLC (5% MeOH in DCM): R*_f_* 0.23; ^1^H NMR (500 MHz, DMSO-*d*_6_): δ 9.60–9.37 (m, 28H), 8.60–8.37 (m, 57H), 8.15 (dq, *J* = 9.6, 5.2 Hz, 19H), 7.94 (ddd, *J* = 13.7, 8.8, 5.5 Hz, 18H), 7.83 (ddd, *J* = 11.7, 7.9, 4.1 Hz, 18H), 7.22–7.10 (m, 14H), 6.94 (s, 2H), 6.70 (s, 28H), 4.20 (d, *J* = 6.4 Hz, 38H), 4.11–3.92 (m, 64H), 3.89–3.67 (m, 37H), 2.91 (d, *J* = 7.6 Hz, 40H), 2.11 (s, 26H), 1.69–1.42 (m, 66H), 1.36–1.10 (m, 310H); ^13^C NMR (126 MHz, DMSO-*d*_6_): δ 206.67, 163.30, 162.78, 159.59, 159.57, 153.94, 153.91, 147.47, 140.77, 140.74, 138.31, 131.76, 130.77, 129.81, 129.04, 128.71, 126.17, 125.14, 124.22, 123.50, 122.95, 101.74, 99.71, 67.79, 64.82, 64.52, 63.93, 61.25, 55.30, 48.13, 35.43, 33.01, 31.72, 31.07, 29.52, 29.40, 29.36, 29.20, 29.14, 29.10, 29.01, 28.95, 28.67, 26.37, 25.94, 25.84, 22.50, 22.13, 14.32, 13.45; MALDI-TOF-MS (DHB matrix) (*m/z*): [M+H]^+^ Calcd. for C_578_H_664_N_94_O_106_ 10624.22; molecular ion was not detected with any matrix. This may be due the fact that the molecule absorbed close to the wavelength of laser, thereby fragmentating the parent ion before it was detected [47]. 

## 4. Conclusions

In summary, polyurethane dendrons to G3 were synthesized using a convergent method, where the azide-alkyne click reaction is highly effective strategy for preparing dendrons with high yields and easy isolation. When two different dendrons reacted with a difunctional core, Janus dendrimers are favorable. This is demonstrated by the yield of the G3 Janus dendrimer, which accounted for more than 90% of the total product yield, showing the promising synthetic applicability of this approach to introduce unsymmetrical branches in the dendritic system with a high efficiency. Photophysical study of these dendrimers showed the FRET phenomenon from naphthalimide (donor) to blue fluorophore (acceptor) resulting in shift of emission toward the longer wavelengths accompanied by an increased intensity. The FRET measurements in the solution showed that red-shifting is not affected by the generation of dendrimers. Our method can prepare dendrimers with fluorophores to achieve large Stokes shifts. This research establishes the parameters for energy transfer within dendritic molecules and should inform the further development in the preparation of molecular antennas.

## Data Availability

The data presented in this study are available upon request from the corresponding author. The data are not publicly available because of the lack of a dedicated server.

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
