# Peer review of "Synthesis of Fluorescent, Dumbbell-Shaped Polyurethane Homo- and Heterodendrimers and Their Photophysical Properties"

_ijms, 2023, doi:10.3390/ijms24021662_

Round 1

Reviewer 1 Report

D.P. Poudel and R.T. Taylor report the synthesis of fluorescent homo- and Janus dendrimers. A comprehensive characterization of new compounds is given, however, more advanced photophysical characterization should be provided before the paper can be accepted for publication.

 1.     The formats of the graphs and tables should be double checked and should fit the manuscript template, not just copy-pasted from Microsoft excel. The world “eflux” in scheme 5 is confusing. Probably, the authors meant “reflux”. There are no references and mentions of figures and information included in Supporting Information throughout the manuscript.

2.     The elemental analysis data on each new compound synthesized should be supplemented along with NMR and mass spectrometry studies.

3.     I recommend to perform GPC analysis for compounds undetected by MALDI ToF (homodendrimers 25, 27). The interactions with matrix will not hamper the analysis in this case. The results allow one to estimate the absence of unsubstituted products and to assess the molecular weight of compounds.

4.     There is only basic description of the photophysical characterization. The authors should present the data on Stokes shifts, fluorescence lifetime and quantum yield of the compounds before the applicability as labeling agents can be stated. Then, the merits and novelty of the manuscript can be stood out.

5.     It is not clear from the data presented, how does the dendrimer generation influence the photophysical properties and does the construction of complicated structures of higher generation dendrimers add the advantages in fluorescence behavior?

6.      Was the shift of emission wavelength being accompanied by the variation in acceptor group emission intensity? This issue should be addressed and discussed in the manuscript for accurate definition of FRET phenomenon.

Reviewer 2 Report

The work entitled "Synthesis of fluorescent, dumbbell-shaped polyurethane homo- and heterodendrimers and their photophysical properties " is written very correctly. All the results are well described in details. I read this paper with great interest and was disappointed at the end. WHAT RESULTS FROM THESE RESEARCH?

The conclusions are not conclusions, but rather a short (in relation to the volume of the article) summary of the research.

Authors must complete their work as follows:

i)                 Abstract:

"Fluorescent dendrimers find wide applications in biomedical and materials science" - Please write at least one sentence – WHAT ARE THE POTENTIAL APPLICATIONS OF THESE MATERIALS?

ii)               Introduction:

At least two sentences should be added - WHY FLUORESCENCE AND ABSORPTION WERE TESTED? What is the impact of the obtained results on the potential applications of these materials?

iii)              Results and discussion:

This point is correct.

        iv) Conclusions:

- Please expand this point and add the CONCLUSIONS, especially what results from photophysical research.

After these corrections, the article can be accepted for publication.

Round 2

Reviewer 1 Report

I recommend to include references in FRET section regarding the examples of FRET phenomenon application.

Author Response

As advised, the references for the FRET applications (ref 40-42) have been inserted.